# Asymptotic Distribution of Certain Types of Entropy under the Multinomial Law

**DOI:** 10.3390/e25050734

**Published:** 2023-04-28

**Authors:** Andrea A. Rey, Alejandro C. Frery, Magdalena Lucini, Juliana Gambini, Eduarda T. C. Chagas, Heitor S. Ramos

**Affiliations:** 1Signal and Image Processing Center, Universidad Tecnológica Nacional, Ciudad Autónoma de Buenos Aires C1179AAQ, Argentina; 2School of Mathematics and Statistics, Victoria University of Wellington, Wellington 6140, New Zealand; 3Department of Mathematics, FaCENA, Universidad Nacional del Nordeste and CONICET, Corrientes W3400AYY, Argentina; 4CIDIA, Universidad Nacional de Hurlingham, Pcia. de Buenos Aires Argentina—CPSI Universidad Tecnológica Nacional, Buenos Aires C1041AAJ, Argentina; 5Departamento de Ciência da Computação, Universidade Federal de Minas Gerais, Belo Horizonte 31270-901, Brazil

**Keywords:** multinomial distribution, entropy, asymptotic distributions, hypothesis tests

## Abstract

**Simple Summary:**

We obtain expressions for the asymptotic distributions of the Rényi and Tsallis of order *q* entropies, and Fisher information when computed on the maximum likelihood estimator of probabilities from multinomial random samples. We recall results related to the Shannon entropy. We build a test for comparing entropies of different types and categories.

**Abstract:**

We obtain expressions for the asymptotic distributions of the Rényi and Tsallis of order *q* entropies and Fisher information when computed on the maximum likelihood estimator of probabilities from multinomial random samples. We verify that these asymptotic models, two of which (Tsallis and Fisher) are normal, describe well a variety of simulated data. In addition, we obtain test statistics for comparing (possibly different types of) entropies from two samples without requiring the same number of categories. Finally, we apply these tests to social survey data and verify that the results are consistent but more general than those obtained with a $\chi^{2}$ test.

## 1. Introduction

The multinomial distribution is an adequate model for describing how observations fall into categories. Quoting Johnson et al. [1], “The Multinomial distribution, like the Multivariate Normal distribution among the continuous multivariate distributions, consumed a sizable amount of the attention that numerous theoretical as well as applied researchers directed towards the area of discrete multivariate distributions.”

The entropy of a (multivariate, in our case) random variable is a substantial quantity. It quantifies the predictability of a system whose outputs can be described by such a model. Entropy has several definitions, both conceptual and mathematical. The concept of entropy originated as a way to relate a system’s energy and temperature [2]. The same concept was used to describe the number of ways the particles of a system can be arranged.

Entropy has been seldom studied as a random variable. Hutcheson [3] and Hutcheson and Shenton [4] discussed the exact expected value and variance of the Shannon entropy under the multinomial model. These works also provided approximate expressions that circumvent the numerical issues when using the exact value.

Jacquet and Szpankowski [5] studied high-quality analytic approximations of the Rényi entropy, of which the Shannon entropy is a particular case, under the binomial model. With the same approach, Cichoń and Golębiewski [6] obtained expressions for more general functionals that include the multinomial distribution. These works treat the entropy as a fixed quantity. Cook et al. [7] studied almost unbiased estimators of functions of the parameter of the binomial distribution. The authors extended those results to find an almost-unbiased estimator for the entropy under multinomial laws.

Chagas et al. [8] treated the Shannon entropy as a random variable. The authors obtained its asymptotic distribution when indexing by the maximum likelihood estimators of the proportions under the multinomial distribution. This result allowed the devising of unilateral and bilateral tests for comparing the entropy from two samples in a very general way. These tests do not require having the same number of categories.

In this work, our attention is directed toward the asymptotic distribution of other forms of entropy under the multinomial model. This allows the comparison of large samples throughout their entropies and, with this, they may have different numbers of classes. The comparison also allows using different types of entropy. We firstly apply the multivariate delta method and, in the case of the Rényi entropy, we transform the resulting multivariate normal distribution into that of the logarithm of the absolute value of a normally distributed random variable. Then, we provide the general expression of a test statistic that suits our needs.

This paper unfolds as follows. Section 2 recalls the main properties of the multinomial distribution and defines the four types of entropies we will study. In Section 3, we present the central results, i.e., the asymptotic distribution of those entropies. We describe the techniques we used and left for Section A.1 technical details. We validate our results with simulation studies in Section 4: we show the adequacy of the normal distribution as limit law for the entropies under three probability models of different support, considering various sample sizes. In Section 5, we show that these asymptotic properties lead to a helpful hypothesis test between samples with different categories. We conclude the article in Section 6. Section A.2 comments on applications that justify our choices of the number of categories and sample sizes in the simulation studies. Section A.3 discloses relevant computational information, including reproducibility.

## 2. Entropies and the Multinomial
Distribution

Consider a series of *n* independent trials, where only one of *k* mutually exclusive events $\pi^{1},\pi^{2},\ldots ,\pi^{k}$ must be observed in each one, with probability $p=\{p_{1},p_{2},\ldots ,p_{k}\}$ such that $p_{\ell }\geq 0$ and $\sum_{\ell =1}^{k}p_{\ell }=1$. Let $N=(N_{1},N_{2},\ldots ,N_{k})$ be the random vector that counts the number of occurrences of the events $\pi^{1},\pi^{2},\ldots ,\pi^{k}$ in the *n* trials, with $N_{\ell }\geq 0$ and $\sum_{\ell =1}^{k}N_{\ell }=n$. A sample from N, say n, is a *k*-variate vector of integer values $n=(n_{1},n_{2},\ldots ,n_{k})$. Then, the joint distribution of N is
(1)$$\mathrm{Pr}\left(N=n\right)=\mathrm{Pr}\left(N_{1}=n_{1},N_{2}=n_{2},\ldots ,N_{k}=n_{k}\right)=n!\prod_{\ell =1}^{k}\frac{p_{\ell }^{n_{\ell }}}{n_{\ell }!}.$$

We denote this situation as N∼Mult(n,p).

In practice, one does not know the true values of p, the probabilities that index this multinomial distribution. Such values are estimated by computing ${\hat{p}}_{\ell }$, the proportion of times the class (category, event) $\pi^{\ell }$ was observed among the *k* possible categories $\pi =\{\pi^{1},\pi^{2},\ldots ,\pi^{k}\}$ during the *n* trials. The maximum likelihood estimator for $\hat{p}=\left({\hat{p}}_{1},{\hat{p}}_{2},\ldots ,{\hat{p}}_{k}\right)$ is the random vector of proportions. This maximum likelihood estimator coincides with the intuitive estimator based on the distribution’s first moments, and is the most frequently used in applications.

We study the distribution of several forms of entropy of the random vector $\hat{p}$ for fixed *k*. Notice that $\hat{p}$ is computed over a single *k*-variate measurement of random proportions corresponding to a single random sample from N∼Mult(n,p). The asymptotic behaviors we derive hold for typical cases in which n≫k.

The Shannon entropy measures the disorder or unpredictability of systems characterized by a probability distribution. On the one hand, the minimum Shannon value occurs when there is complete knowledge about the system behavior and total confidence in predicting the following observation. On the other hand, when a uniform distribution describes the system’s behavior, that is, when all possibilities have the same probability of occurrence, the knowledge about the behavior of the data is minimal. In Chagas et al. [8], we studied the asymptotic distribution of the Shannon entropy. In this work, we extend those results to three other forms of entropy.

Other types of descriptors have been proposed in the literature to extract additional information not captured by the Shannon entropy. Tsallis [9] and Rényi [10], for instance, proposed parametric versions, which include the Shannon entropy.

Fisher information [11] is defined by an average logarithm derivative of a continuous probability density function. In the case of discrete densities, this measure can be approximated using differences of probabilities between consecutive distribution elements. While the Shannon entropy captures the degree of unpredictability of a system, the Fisher information is related to the rate of change of consecutive observations and, thus, quantifies small changes and perturbations.

Given a type of entropy *H*, we are interested in the distribution of H(p) when indexed by $\hat{p}$, the maximum likelihood estimator of p. Our problem then becomes finding the distribution of $H(\hat{p})$ for the following:The Shannon entropy
(2)$$H_{S}\left(\hat{p}\right)=-\sum_{\ell =1}^{k}{\hat{p}}_{\ell }\log{\hat{p}}_{\ell },$$The Tsallis entropy with index q∈R\{1}
(3)$$H_{T}^{q}\left(\hat{p}\right)=\sum_{\ell =1}^{k}\frac{{\hat{p}}_{\ell }-{\hat{p}}_{\ell }^{q}}{q-1},$$The Rényi entropy of order $q\in R^{+}\backslash \left\{1\right\}$
(4)$$H_{R}^{q}\left(\hat{p}\right)=\frac{1}{1-q}\log\sum_{\ell =1}^{k}{\hat{p}}_{\ell }^{q},$$The Fisher information, also termed “Fisher Information Measure” in the literature, with renormalization coefficient $F_{0}=4$
(5)$$H_{F}\left(\hat{p}\right)=F_{0}\sum_{\ell =1}^{k-1}{\left(\sqrt{{\hat{p}}_{\ell +1}}-\sqrt{{\hat{p}}_{\ell }}\right)}^{2}.$$Among other possibilities, we used Equation (2.7) from Ref. [12].

## 3. Asymptotic Distributions of Entropies

The main results of this section are the asymptotic distributions of the Shannon (Equation 2), Tsallis of order *q* (Equation 3), and Rényi of order *q* (Equation 4) entropies, and Fisher information (Equation 5). These results are presented, respectively, in Equations (Equation 30)–(Equation 32) and (Equation 35). Notice that the Rényi entropy is not asymptotically normally distributed, while the other three are.

We recall the following theorems known respectively as the delta method and its multivariate version. We refer to Lehmann and Casella [13] for their proofs.

**Theorem** **1.**
*Let $X_{n}$ be a sequence of independent and identically distributed random variables such that $\sqrt{n}\left[X_{n}-\theta \right]$ converges in distribution to a $N(0,\sigma^{2})$. If ∂h/∂θ exists and does not vanish, then $\sqrt{n}\left[h\left(X_{n}\right)-h\left(\theta \right)\right]$ converges in distribution to a $N(0,\sigma^{2}{\left[\partial h/\partial \theta \right]}^{2})$.*


**Theorem** **2.**
*Let $X_{n}=\left(X_{1n},X_{2n},\ldots ,X_{kn}\right)$ be a sequence of independent and identically distributed vectors of random variables such that $\sqrt{n}\left[X_{1n}-\theta_{1},X_{2n}-\theta_{2},\ldots ,X_{kn}-\theta_{k}\right]$ converges in distribution to a multivariate normal distribution $N_{n}\left(0,\Sigma \right)$, where Σ is the covariance matrix. Suppose that $h_{1},h_{2},\ldots ,h_{k}$ are real functions continuously differentiable in a neighborhood of the parameter point $\theta =(\theta_{1},\theta_{2},\ldots ,\theta_{k})$ and such that the matrix of partial derivatives $B={\left(\partial h_{\ell }/\partial \theta_{j}\right)}_{\ell ,j=1}^{k}$ is non-singular in the mentioned neighborhood. Then, the following convergence in distribution holds:*

$$\sqrt{n}\left[h_{1}\left(X_{n}\right)-h_{1}\left(\theta \right),h_{2}\left(X_{n}\right)-h_{2}\left(\theta \right),\ldots ,h_{k}\left(X_{n}\right)-h_{k}\left(\theta \right)\right]\overset{D}{\to }N\left(0,B\Sigma B^{\prime }\right),$$

*where $B^{\prime }$ denotes the transpose of B.*


Now, we focus on the case N∼Mult(n,p). Let $\hat{p}=N/n$ be the vector of sample proportions which coincides with the maximum likelihood estimator (MLE) of p and $Y_{n}=\sqrt{n}\left(\hat{p}-p\right)$. Then
$$Y_{n}\overset{D}{\to }N\left(0,D_{p}-pp^{\prime }\right),$$
where $D_{p}=\mathrm{Diag}\left(p_{1},p_{2},\ldots ,p_{k}\right)$.

Let us explore the covariance matrix in this case:(6)$$\begin{array}{cc} D_{p}-pp^{\prime } & =\left(\begin{array}{cccc} p_{1} & 0 & \cdots & 0\\ 0 & p_{2} & \cdots & 0\\ \vdots & \vdots & \ddots & \vdots \\ 0 & 0 & \cdots & p_{k} \end{array}\right)-\left(\begin{array}{c} p_{1}\\ p_{2}\\ \vdots \\ p_{k} \end{array}\right)\left(\begin{array}{cccc} p_{1} & p_{2} & \cdots & p_{k} \end{array}\right) \end{array}$$(7)$$\begin{array}{cc} & =\left(\begin{array}{cccc} p_{1}-p_{1}^{2} & -p_{1}p_{2} & \cdots & -p_{1}p_{k}\\ -p_{2}p_{1} & p_{2}-p_{2}^{2} & \cdots & -p_{2}p_{k}\\ \vdots & \vdots & \ddots & \vdots \\ -p_{k}p_{1} & -p_{k}p_{2} & \cdots & p_{k}-p_{k}^{2} \end{array}\right) \end{array}$$

It means that the covariance matrix $\Sigma_{p}\in R^{k\times k}$ we are interested in is of the form
(8)$${\left(\Sigma_{p}\right)}_{\ell j}=\left\{\begin{array}{cc} p_{\ell }\left(1-p_{\ell }\right) & \mathrm{if}\;\ell =j,\\ -p_{\ell }p_{j} & \mathrm{if}\;\ell \neq j. \end{array}\right.$$

The above statements are generalized. In the following, we obtain new results for the Tsallis and Rényi entropies, and for the Fisher information. For the sake of completeness, we also include the results for the Shannon entropy.

In order to apply the delta method using Theorem 2, we consider the following functions: (9)$$\begin{array}{cc} h_{\ell }^{S}\left(p_{1},p_{2},\ldots ,p_{k}\right) & =p_{\ell }\log p_{\ell }, \end{array}$$(10)$$\begin{array}{cc} h_{\ell }^{T}\left(p_{1},p_{2},\ldots ,p_{k}\right) & =p_{\ell }-p_{\ell }^{q}, \end{array}$$(11)$$\begin{array}{cc} h_{\ell }^{R}\left(p_{1},p_{2},\ldots ,p_{k}\right) & =p_{\ell }^{q}, \end{array}$$(12)$$\begin{array}{cc} h_{\ell }^{F}\left(p_{1},p_{2},\ldots ,p_{k}\right) & ={\left(\sqrt{p_{\ell +1}}-\sqrt{p_{\ell }}\right)}^{2}, \end{array}$$
for ℓ=1,2,…,k except for () that holds for ℓ=1,2,…,k−1. The assumptions are verified, and thus,
(13)$$\begin{array}{cc} \; & \frac{\partial h_{\ell }^{S}}{\partial p_{\ell }}=\log p_{\ell }+1\;\mathrm{and}\;\frac{\partial h_{\ell }^{S}}{\partial p_{j}}=0\;\mathrm{if}\;j\neq \ell , \end{array}$$
(14)$$\begin{array}{cc} \; & \frac{\partial h_{\ell }^{T}}{\partial p_{\ell }}=1-qp_{\ell }^{q-1}\;\mathrm{and}\;\frac{\partial h_{\ell }^{T}}{\partial p_{j}}=0\;\mathrm{if}\;j\neq \ell , \end{array}$$
(15)$$\begin{array}{cc} \; & \frac{\partial h_{\ell }^{R}}{\partial p_{\ell }}=qp_{\ell }^{q-1}\;\mathrm{and}\;\frac{\partial h_{\ell }^{R}}{\partial p_{j}}=0\;\mathrm{if}\;j\neq \ell , \end{array}$$
(16)$$\begin{array}{cc} \; & \frac{\partial h_{\ell }^{F}}{\partial p_{j}}=\frac{\sqrt{p_{\ell +1}}-\sqrt{p_{\ell }}}{{\left(-1\right)}^{\ell +j-1}\sqrt{p_{j}}}\;\mathrm{if}\;j=\ell ,\ell +1\;\mathrm{and}\;\frac{\partial h_{\ell }^{F}}{\partial p_{j}}=0\;\mathrm{if}\;j\neq \ell ,\ell +1. \end{array}$$

Finally, we need the covariance matrix of the multivariate normal limit distribution, which is
(17)$$\Sigma_{p}^{\Delta M}={\left(\frac{\partial h_{\ell }^{M}}{\partial p_{j}}\right)}_{\ell ,j=1}^{k}\Sigma_{p}{{\left(\frac{\partial h_{\ell }^{M}}{\partial p_{j}}\right)}_{\ell ,j=1}^{k}}^{\prime },$$
where M∈{S,T,R,F}. Since ${\left(\partial h_{\ell }^{M}/\partial p_{j}\right)}_{\ell ,j=1}^{k}$ are diagonal matrices for M∈{S,T,R}, we can use Equation (Equation 38) to conclude that
(18)$$\begin{array}{cc} \; & {\left(\Sigma_{p}^{\Delta S}\right)}_{\ell j}=\left\{\begin{array}{cc} \left(p_{\ell }-p_{\ell }^{2}\right){\left(\log p_{\ell }+1\right)}^{2} & \mathrm{if}\;\ell =j,\\ -p_{\ell }p_{j}\left(\log p_{\ell }+1\right)\left(\log p_{j}+1\right) & \mathrm{if}\;\ell \neq j; \end{array}\right. \end{array}$$
(19)$$\begin{array}{cc} \; & {\left(\Sigma_{p}^{\Delta T}\right)}_{\ell j}=\left\{\begin{array}{cc} \left(p_{\ell }-p_{\ell }^{2}\right){\left(1-qp_{\ell }^{q-1}\right)}^{2} & \mathrm{if}\;\ell =j,\\ -p_{\ell }p_{j}\left(1-qp_{\ell }^{q-1}\right)\left(1-qp_{j}^{q-1}\right) & \mathrm{if}\;\ell \neq j; \end{array}\right. \end{array}$$
(20)$$\begin{array}{cc} \; & {\left(\Sigma_{p}^{\Delta R}\right)}_{\ell j}=\left\{\begin{array}{cc} q^{2}\left(p_{\ell }-p_{\ell }^{2}\right)p_{\ell }^{2(q-1)} & \mathrm{if}\;\ell =j,\\ -q^{2}{\left(p_{\ell }p_{j}\right)}^{q} & \mathrm{if}\;\ell \neq j. \end{array}\right. \end{array}$$

In the case of $\Sigma_{p}^{\Delta F}$, from Equations (Equation 40) and (Equation 41) we have the following:For ℓ,j=1,2,…,k−2 and ℓ≠j−1,j,j+1:
(21)$${\left(\Sigma_{p}^{\Delta F}\right)}_{\ell j}=\left(\sqrt{p_{\ell +1}}-\sqrt{p_{\ell }}\right)\left(\sqrt{p_{j+1}}-\sqrt{p_{j}}\right)\left(\sqrt{p_{\ell +1}p_{j}}+\sqrt{p_{\ell }p_{j+1}}-\sqrt{p_{\ell }p_{j}}-\sqrt{p_{\ell +1}p_{j+1}}\right).$$For ℓ=1,2,…,k−2:
(22)$$\begin{array}{cc} {\left(\Sigma_{p}^{\Delta F}\right)}_{\ell ,\ell -1}=\left(\sqrt{p_{\ell +1}}-\sqrt{p_{\ell }}\right) & \left(\sqrt{p_{\ell }}-\sqrt{p_{\ell -1}}\right)\\ & \left(\sqrt{p_{\ell +1}p_{\ell -1}}+p_{\ell }-1-\sqrt{p_{\ell }p_{\ell -1}}-\sqrt{p_{\ell +1}p_{\ell }}\right). \end{array}$$For ℓ=1,2,…,k−2:
(23)$${\left(\Sigma_{p}^{\Delta F}\right)}_{\ell \ell }={\left(\sqrt{p_{\ell +1}}-\sqrt{p_{\ell }}\right)}^{2}\left(2\sqrt{p_{\ell }p_{\ell +1}}+2-p_{\ell }-p_{\ell +1}\right).$$For ℓ=1,2,…,k−2:
(24)$${\left(\Sigma_{p}^{\Delta F}\right)}_{\ell ,\ell +1}=\left(\sqrt{p_{\ell +1}}-\sqrt{p_{\ell }}\right)\left(\sqrt{p_{\ell +2}}-\sqrt{p_{\ell +1}}\right)\left(p_{\ell +1}-1+\sqrt{p_{\ell }p_{\ell +2}}-\sqrt{p_{\ell }p_{\ell +1}}-\sqrt{p_{\ell +1}p_{\ell +2}}\right).$$For j=1,2,…,k−2:
(25)$${\left(\Sigma_{p}^{\Delta F}\right)}_{k-1,j}=\left(\sqrt{p_{k}}-\sqrt{p_{k-1}}\right)\left(\sqrt{p_{j+1}}-\sqrt{p_{j}}\right)\left(\frac{p_{k}p_{j+1}}{\sqrt{p_{j+1}}}-\frac{p_{k-1}p_{j}}{\sqrt{p_{j}}}\right).$$Finally,
(26)$${\left(\Sigma_{p}^{\Delta F}\right)}_{k-1,k-1}={\left(\sqrt{p_{k}}-\sqrt{p_{k-1}}\right)}^{2}\left(1-p_{k-1}\right).$$

Hence, we conclude that
(27)$$\sqrt{n}\left[h_{1}^{M}\left({\hat{p}}_{1}\right)-h_{1}^{M}\left(p_{1}\right),h_{2}^{M}\left({\hat{p}}_{2}\right)-h_{2}^{M}\left(p_{2}\right),\ldots ,h_{k^{\prime }}^{M}{\hat{p}}_{k^{\prime }}-h_{k^{\prime }}^{M}\left(p_{k^{\prime }}\right)\right]\overset{D}{\to }N\left(0,\Sigma_{p}^{\Delta M}\right),$$
where M∈{S,T,R,F} and $k^{\prime }=k$ in all cases except for the case of the Fisher information in which $k^{\prime }=k-1$. An equivalent expression is
(28)$$\sqrt{n}\left[h_{1}^{M}\left({\hat{p}}_{1}\right),h_{2}^{M}\left({\hat{p}}_{2}\right),\ldots ,h_{k^{\prime }}^{M}\left({\hat{p}}_{k^{\prime }}\right)\right]\overset{D}{\to }N\left(\sqrt{n}\left(\begin{array}{c} h_{1}^{M}\left(p_{1}\right)\\ h_{2}^{M}\left(p_{2}\right)\\ \vdots \\ h_{k^{\prime }}^{M}\left(p_{k^{\prime }}\right) \end{array}\right),\Sigma_{p}^{\Delta M}\right).$$

If Y is a vector of random variables such that $\sqrt{n}Y\overset{D}{\to }N\left(\sqrt{n}\mu ,\Sigma \right)$, then it can be proved that $E\left(\sqrt{n}Y\right)\to \sqrt{n}\mu$ and $\mathrm{Var}(\sqrt{n}Y)\to \Sigma$. Provided well-known properties, it holds that E(Y)→μ and Var(Y)→1/nΣ. Applying this to (Equation 28),
(29)$$\left[h_{1}^{M}\left({\hat{p}}_{1}\right),h_{2}^{M}\left({\hat{p}}_{2}\right),\ldots ,h_{k^{\prime }}^{M}\left({\hat{p}}_{k^{\prime }}\right)\right]\overset{D}{\to }N\left(\left(\begin{array}{c} h_{1}^{M}\left(p_{1}\right)\\ h_{2}^{M}\left(p_{2}\right)\\ \vdots \\ h_{k^{\prime }}^{M}\left(p_{k^{\prime }}\right) \end{array}\right),\frac{1}{n}\Sigma_{p}^{\Delta M}\right).$$

Now, using (Equation 29), we find the asymptotic distribution of (Equation 2)–(Equation 5). In order to do so, we need to know the distribution of the sum of *k* Gaussian random variables with different means and an arbitrary covariance matrix.

For any *k*-dimensional multivariate normal distribution Z∼N(μ,Σ), with $\mu \in R^{k}$ and covariance matrix $\Sigma =(\sigma_{\ell j})$, holds that the distribution of $W=a^{T}Z$, with $a\in R^{k}$, is $N\left(a^{T}\mu ,\sum_{\ell =1}^{k}a_{\ell }^{2}\sigma_{\ell \ell }+2\sum_{\ell =1}^{k-1}\sum_{j=i+1}^{k}a_{\ell }a_{j}\sigma_{\ell j}\right)$. Using the limit distribution presented in (Equation 29) and a=(−1,−1,…,−1), we directly have the asymptotic distribution of the Shannon entropy as follows:(30)$$\begin{array}{c} H_{S}\left(\hat{p}\right)=-\sum_{\ell =1}^{k}{\hat{p}}_{\ell }\log{\hat{p}}_{\ell }\overset{D}{\to }\\ N\left(-\sum_{\ell =1}^{k}p_{\ell }\log p_{\ell },\frac{1}{n}\sum_{\ell =1}^{k}p_{\ell }\left(1-p_{\ell }\right){\left(\log p_{\ell }+1\right)}^{2}-\frac{2}{n}\sum_{j=1}^{k-1}\sum_{\ell =j+1}^{k}p_{\ell }p_{j}\left(\log p_{\ell }+1\right)\left(\log p_{j}+1\right)\right). \end{array}$$

With similar arguments and a=(1,1,…,1), we obtain the asymptotic distribution for the Tsallis entropy of order *q*:(31)$$H_{T}^{q}\left(\hat{p}\right)=\sum_{\ell =1}^{k}\frac{{\hat{p}}_{\ell }-{\hat{p}}_{\ell }^{q}}{q-1}\overset{D}{\to }N\left(\sum_{\ell =1}^{k}\frac{p_{\ell }-p_{\ell }^{q}}{q-1},\sum_{\ell =1}^{k}\frac{\left(p_{\ell }-p_{\ell }^{2}\right){\left(1-qp_{\ell }^{q-1}\right)}^{2}}{n{\left(q-1\right)}^{2}}-2\sum_{j=1}^{k-1}\sum_{\ell =j+1}^{k}\frac{p_{\ell }p_{j}\left(1-qp_{\ell }^{q-1}\right)\left(1-qp_{j}^{q-1}\right)}{n{\left(q-1\right)}^{2}}\right).$$

The procedure is analogous for the Fisher information but with $a=\left(1,1,\ldots ,1\right)\in R^{k-1}$. Hence, it can be proved that
(32)$$H_{F}\left(\hat{p}\right)=F_{0}\sum_{\ell =1}^{k-1}{\left(\sqrt{{\hat{p}}_{\ell -1}}-\sqrt{{\hat{p}}_{\ell }}\right)}^{2}\overset{D}{\to }N\left(F_{0}\sum_{\ell =1}^{k-1}{\left(\sqrt{p_{\ell -1}}-\sqrt{p_{\ell }}\right)}^{2},\frac{F_{0}}{n}\Sigma^{*}\right),$$
where
(33)$$\begin{array}{c} \Sigma^{*}={\left(\sqrt{p_{k}}-\sqrt{p_{k-1}}\right)}^{2}\left(1-p_{k-1}\right)+\sum_{\ell =1}^{k-2}{\left(\sqrt{p_{\ell +1}}-\sqrt{p_{\ell }}\right)}^{2}\left(2\sqrt{p_{\ell }p_{\ell +1}}-p_{\ell }-p_{\ell +1}+2\right)+\\ 2\sum_{\ell =3}^{k-2}\sum_{j=1}^{\ell -2}\left[\left(\sqrt{p_{\ell +1}}-\sqrt{p_{\ell }}\right)\left(\sqrt{p_{j+1}}-\sqrt{p_{j}}\right)\left(\sqrt{p_{\ell +1}p_{j}}+\sqrt{p_{\ell }p_{j+1}}-\sqrt{p_{\ell }p_{j}}-\sqrt{p_{\ell +1}p_{j+1}}\right)\right]+\\ 2\sum_{j=1}^{k-2}\left[\left(\sqrt{p_{k}}-\sqrt{p_{k-1}}\right)\left(\sqrt{p_{j+1}}-\sqrt{p_{j}}\right)\left(p_{k}\sqrt{p_{j+1}}-p_{k-1}\sqrt{p_{j}}\right)\right]+\\ 2\sum_{\ell =2}^{k-2}\left[\left(\sqrt{p_{\ell +1}}-\sqrt{p_{\ell }}\right)\left(\sqrt{p_{\ell }}-\sqrt{p_{\ell -1}}\right)\left(\sqrt{p_{\ell +1}p_{\ell -1}}-\sqrt{p_{\ell }p_{\ell -1}}-\sqrt{p_{\ell +1}p_{\ell }}+p_{\ell }-1\right)\right]. \end{array}$$

To obtain expression (Equation 33), we use the symmetry of the covariance matrix which implies that $\sum_{\ell =1}^{k-1}\sum_{j=\ell +1}^{k}a_{\ell }a_{j}\sigma_{\ell j}=\sum_{\ell =2}^{k-1}\sum_{j=1}^{\ell -1}a_{\ell }a_{j}\sigma_{\ell j}$. It is worth noticing that the expression of the covariance matrix for Fisher information is more complicated than the previously analyzed entropies since the matrix of partial derivatives is not diagonal in this case.

The case of Rényi entropy is different because, following the previous methodology, we can prove that
(34)$$\sum_{\ell =1}^{k}{\hat{p}}_{\ell }^{q}\overset{D}{\to }N\left(\sum_{\ell =1}^{k}p_{\ell }^{q},\frac{1}{n}\sum_{\ell =1}^{k}q^{2}\left(p_{\ell }-p_{\ell }^{2}\right)p_{\ell }^{2(q-1)}-\frac{2}{n}\sum_{\ell =1}^{k-1}\sum_{j=\ell +1}^{k}q^{2}{\left(p_{j}p_{\ell }\right)}^{q}\right).$$

Hence,
(35)$$H_{R}^{q}\left(\hat{p}\right)=\frac{1}{1-q}\log\sum_{\ell =1}^{k}{\hat{p}}_{\ell }^{q}\overset{D}{\to }P_{R}^{q},$$
where
(36)$$P_{R}^{q}\left(x\right)=\frac{1-q}{\sigma^{*}\sqrt{2\pi }}\exp\left[\left(1-q\right)x\log\left(k\right)\right]\exp\left\{-\frac{1}{2}{\left(\frac{\exp\left[\left(1-q\right)x\log\left(k\right)\right]-\mu^{*}}{\sigma^{*}}\right)}^{2}\right\},$$
with $\mu^{*}=\sum_{\ell =1}^{k}p_{\ell }^{q}$ and $\sigma^{*}=n^{-1}\sum_{\ell =1}^{k}q^{2}\left(p_{\ell }-p_{\ell }^{2}\right)p_{\ell }^{2(q-1)}-2n^{-1}\sum_{\ell =1}^{k-1}\sum_{j=\ell +1}^{k}q^{2}{\left(p_{\ell }p_{j}\right)}^{q}$. Notice that this is not a normal distribution but the distribution of the logarithm of the absolute value of a normally distributed random variable.

Often, in practice, these entropies are scaled to be in [0,1]; these are called “normalized entropies”. The following modifications must be considered in the normalized versions of the entropies. For the normalized Shannon entropy, the asymptotic mean and variance are multiplied by 1/logk and $1/{\left(\log k\right)}^{2}$, respectively. In the case of the normalized Tsallis entropy, the asymptotic mean and variance are multiplied by $\left(q-1\right)/\left(1-k^{1-q}\right)$ and ${\left(q-1\right)}^{2}/{\left(1-k^{1-q}\right)}^{2}$, respectively. Finally, the asymptotic distribution of the normalized Rényi entropy is ${\tilde{P}}_{R}^{q}\left(x\right)=\log kP_{R}^{q}\left(x\log k\right)$. Notice that normalized entropies do not depend on the logarithm basis. The Fisher information is, as defined in (Equation 5), already normalized.

## 4. Analysis and Validation

In this section, we study the empirical distribution of the entropies computed from $\hat{p}$ under three models, four categories (k∈{6,24,120,720}), and three sample sizes ($n\in \{10^{2}k,10^{3}k,10^{4}k\}$) that depend on the number of categories. These choices of *k* and *n* are based on the values that appear in signal analysis with ordinal patterns; see details of this technique in Section A.2.

We considered the following probability functions $p=(p_{1},p_{2},\ldots ,p_{k})$:Linear: $p_{\ell }=2\ell /\left(k\left(k+1\right)\right)$, 1≤ℓ≤k.One-Almost-Zero: $p_{\ell }=1/k$ for 1≤ℓ≤k−2, $p_{k-1}=\epsilon_{0}$, and $p_{k}=2/k-\epsilon_{0}$ with $\epsilon_{0}=2.220446\times 10^{-16}$ (the smallest positive number for which, in our computer platform, $1+\epsilon_{0}>1$).Half-and-Half: $p_{\ell }=1/k+\epsilon /k$ for 1≤ℓ≤k/2, and $p_{\ell }=1/k-\epsilon /k$ for k/2+1≤ℓ≤k, with ϵ∈{0.1,0.3,0.5,0.8}.

These probability functions are illustrated, for k=6 and ϵ=0.3, in Figure 1. We studied the behavior of the Shannon entropy, the Rényi entropy with q∈{1/3,2/5}, the Tsallis entropy with q∈{1/2,3/2}, and the Fisher information computed on samples of sizes $n\in \{10^{2}k,10^{3}k,10^{4}k\}$. We used 300 independent samples (replicates).

Although Equation (Equation 35) shows that the Rényi entropy is not asymptotically normal, we verified that its density is similar to that of a Gaussian distribution. With this in mind, we also checked of the normality of Rényi entropies. We used the Anderson–Darling test to verify the null hypothesis that the data follow a normal distribution. We chose this test because it uses the hypothesized distribution in calculating critical values. This test is more sensitive than other alternatives; see, for instance, the book by Lehman and Romano [14].

From Table 1, we notice that the Fisher information is the one that fails most times to pass the normality test at the 1% The situation that appears with p-value=0.0010 in the table has, in fact, $p-\mathrm{value}=9.606130\times 10^{-3}$; the table shows rounded values. Figure 2 shows four of these cases, namely for k=6, n=600, and ϵ=0.1,0.3,0.5,0.8. We notice that the deviation from the normal hypothesis is more prevalent in both tails, being that the observations are larger than the theoretical quantiles.

The normality hypothesis was rejected at the 1% level by the Anderson–Darling test in only 24 out of 432 situations, showing that the asymptotic Gaussian model for the entropies is a good description for these data. Table 1 shows those situations.

With the aim to assess the goodness of fit of the asymptotic models, we applied the Kolmogorov–Smirnov test to fifty replicates of samples. Table 2 shows the results where the *p*-value of the test is at least equal to 0.05.

It is worth noticing that even in those cases where the *p*-value is lesser than 0.05, the asymptotic models are a good fit to the data as can be seen in several examples exhibited in Figure 3. The Fisher information shows the worst fitting. Additionally, notice in Figure 3d that, although the asymptotic distribution of the Rényi entropy is not normal, the probability density function is visually very close to the Gaussian model. We verified this similarity in all the cases we considered.

## 5. Application

Inspired by an example from Agresti [16] (p. 200), we extracted data from the General Social Survey (GSS, a project of the independent research organization NORC at the University of Chicago, with principal funding from the National Science Foundation, available at https://gss.norc.org/. The data were downloaded on 24 December 2022). Table 3 shows the level of agreement to the assertion “Religious people are often too intolerant” as measured in three years.

The *p*-values of pairwise $\chi^{2}$ tests for the null hypotheses that the underlying probabilities are equal are

**1998 and 2008:**$3.43\times 10^{-22}$,**1998 and 2018:**$2.01\times 10^{-8}$,**2008 and 2018:**$1.06\times 10^{-3}$.

On the one hand, these values attest that 1998 and 2008 and 1998 and 2018 are very different. On the other hand, although significant, the change between 2008 and 2018 is not so significant.

Table 4 shows the asymptotic mean and variance (in entropies normalized units) of the entropies of the proportions reported in Table 3.

We perform the same hypothesis test with the asymptotic quantities presented in Table 4. Table 5 shows the *p*-values of the null hypothesis that the entropies are equal, using the test discussed by Chagas et al. [8] (Section 5):(37)$$p-\mathrm{value}\approx 2\left(1-\Phi \left(\frac{\left|H\left(\hat{p_{1}}\right)-H\left(\hat{p_{2}}\right)\right|}{\sqrt{{\hat{\sigma^{2}}}_{n_{1},\hat{p_{1}}}+{\hat{\sigma^{2}}}_{n_{2},\hat{p_{2}}}}}\right)\right),$$
where Φ is the cumulative distribution function of a standard normal random variable, *H* is any of the considered entropies computed with the observed proportions $\hat{p_{i}}$, i=1,2, and ${\hat{\sigma^{2}}}_{n_{i},\hat{p_{i}}}$ is the corresponding sample asymptotic variance that takes into account the sample size $n_{i}$. Notice that the test based on entropies compares only these features, and not the underlying distribution.

The results in Table 5 are consistent with those provided by the $\chi^{2}$ tests, i.e., the most significant differences arise between 1998 and 2008 and between 1998 and 2018. Moreover, the tests based on entropies do not reject the null hypothesis in the pair 2008–2018, except for Rényi entropy of order 2/3. The increased *p*-values are a consequence of the information reduction: whereas the $\chi^{2}$ test compares count-by-count, those based on entropies compare two scalars.

In the second part of this application, we will illustrate the use of test statistics based on entropies for comparing samples with different categories. Situations like this may appear when applying alternative versions of the same questionnaire in a series of surveys.

We collapsed the categories of 1998 into three: “agreement” (by adding “strongly agree” and “agree”), “indifference” (“not agree/disagree”), and “disagreement” (by adding “disagree” and “strong disagree”). The resulting asymptotic mean entropies and asymptotic variances are shown in Table 6.

Table 7 presents the *p*-values of the tests that verify the null hypothesis of the same entropy between the collapsed 1998 data (three categories), and 2008 and 2018 (five categories). These results agree with those presented in Table 5. Such an agreement suggests that, although the number of categories was reduced in 1998 from five to three, the tests based on entropies cope with the loss of information.

## 6. Conclusions

We presented expressions for the asymptotic distribution of the Rényi and Tsallis entropies of order *q*, and Fisher information. The Fisher information and the Tsallis and Shannon entropies have limit normal distribution with means and variances that depend on the underlying probability of patterns and the number of patterns. The Rényi entropy follows, asymptotically, a different distribution, cf. (Equation 35), but a Gaussian law can well approximate it. Those expressions pose no numerical challenges other than setting 0log0≐0. We verified that these asymptotic distributions are good models for data arising from both simulations with a variety of models and from the analysis of actual data.

On the one hand, the Fisher information is the one that fails more frequently to pass the Anderson–Darling normality tests. On the other hand, it does not provide evidence to reject the same hypothesis under the One-Almost-Zero model.

The distributions we present here can be used for building test statistics, as discussed by Chagas et al. [8]. Moreover, Equation (Equation 37) allows performing tests with mixed types of distributions, a situation that may appear in Internet of Things applications, in which, citing Borges et al. [17], one has to deal with “large time series data generated at different rates, of different types and magnitudes, possibly having issues concerning uncertainty, inconsistency, and incompleteness due to missing readings and sensor failures.”

## Figures and Tables

**Figure 1 entropy-25-00734-f001:**
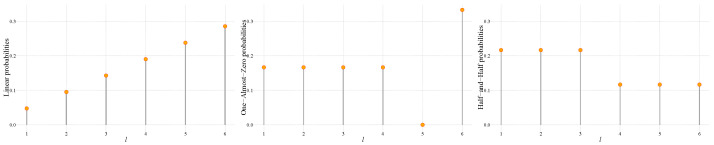
Linear, One-Almost-Zero, and Half-and-Half probability functions for k=6 and ϵ=0.3.

**Figure 2 entropy-25-00734-f002:**
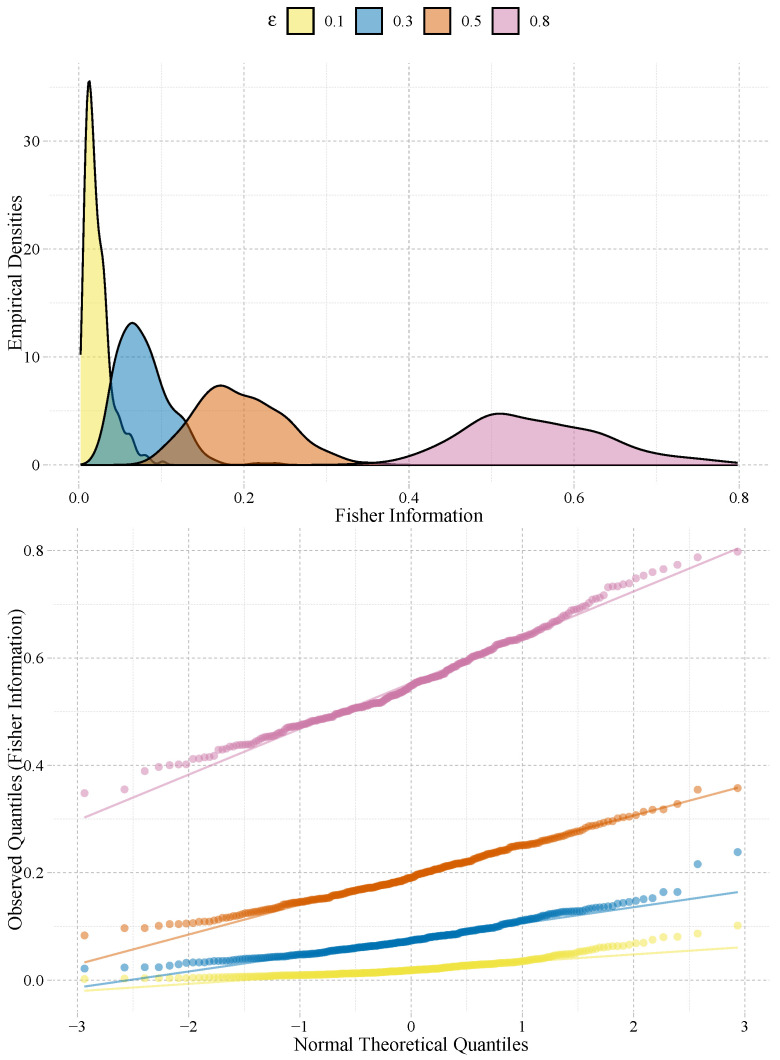
Empirical densities and normal QQ-plots of the Fisher information in situations that fail to pass the normality test at 1%.

**Figure 3 entropy-25-00734-f003:**
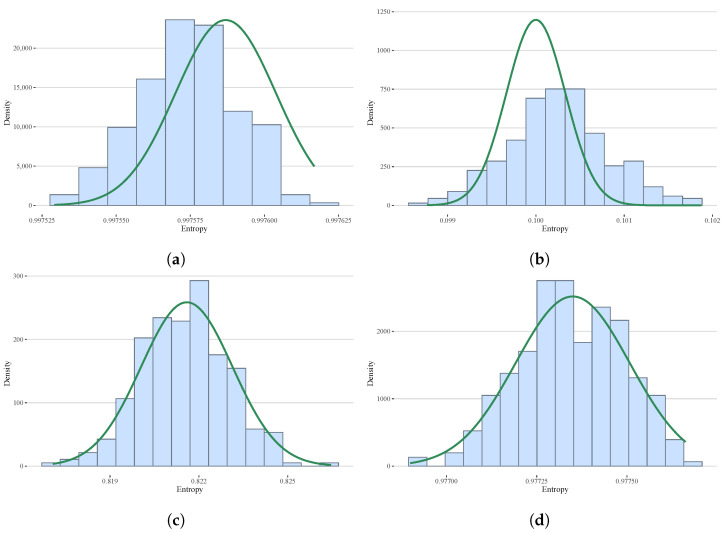
Examples of cases where the null hypothesis of the Kolmogorov–Smirnov test is rejected. The histograms are computed with samples of size 300 using the Freedman–Diaconis rule [15], and the green lines are the asymptotic probability density functions. (**a**) Type: $H_{S}$, Model: OAZ, k=120, $n=10^{4}k$, p-val=0.00202; (**b**) Type: $H_{F}$, Model: OAZ, k=120, $n=10^{4}k$, p-val=0.00013; (**c**) Type: $H_{R}^{1/3}$, Model: Linear, k=24, $n=10^{4}k$, p-val≈0; (**d**) Type: $H_{T}^{1/2}$, Model: HaH, ϵ=0.8, k=6, $n=10^{4}k$, p-val=0.04297.

**Table 1 entropy-25-00734-t001:** Situations for which the *p*-values of the Anderson–Darling test for the normality of samples of size 300 are less than 0.01 (“HF” stands for the Fisher information; “HaH” and “OAZ” are the Half-And-Half and One-Almost-Zero models).

Type	Model	ϵ	*k*	*n*	*p*-Value
$H_{F}$	HaH	0.8	6	600	0.0030
$H_{F}$	HaH	0.1	6	600	0.0000
$H_{F}$	HaH	0.1	24	2400	0.0000
$H_{F}$	HaH	0.8	24	2400	0.0064
$H_{F}$	HaH	0.1	6	6000	0.0000
$H_{F}$	HaH	0.3	6	600	0.0000
$H_{F}$	HaH	0.1	120	120,000	0.0089
$H_{F}$	HaH	0.1	24	24000	0.0004
$H_{F}$	Linear	0	6	600	0.0000
$H_{F}$	Linear	0	24	2400	0.0000
$H_{R}^{1/3}$	HaH	0.1	6	600	0.0000
$H_{R}^{1/3}$	HaH	0.1	24	2400	0.0001
$H_{R}^{1/3}$	OAZ	0	24	2400	0.0000
$H_{R}^{2/3}$	HaH	0.1	6	600	0.0000
$H_{R}^{2/3}$	HaH	0.1	24	2400	0.0001
$H_{R}^{2/3}$	OAZ	0	24	2400	0.0000
$H_{S}$	HaH	0.1	6	600	0.0001
$H_{S}$	HaH	0.1	24	2400	0.0002
$H_{S}$	OAZ	0	24	2400	0.0000
$H_{T}^{1/2}$	HaH	0.1	6	600	0.0000
$H_{T}^{1/2}$	HaH	0.1	24	2400	0.0001
$H_{T}^{1/2}$	OAZ	0	24	2400	0.0000
$H_{T}^{3/2}$	HaH	0.1	6	600	0.0001
$H_{T}^{3/2}$	HaH	0.1	24	2400	0.0003
$H_{T}^{3/2}$	OAZ	0	24	2400	0.0000

**Table 2 entropy-25-00734-t002:** Situations for which the *p*-values of the Kolmogorov–Smirnov test of samples of size 50 are larger than or equal to 0.05 (“HaH” and “OAZ” are the Half-And-Half and One-Almost-Zero models).

Type	Model	ϵ	*k*	*n*		Type	Model	ϵ	*k*	*n*
$H_{S}$	HaH	0.1	6,24	$10^{3}k,10^{4}k$		$H_{T}^{3/2}$	HaH	0.1	6,24	$10^{3}k,10^{4}k$
			120,720	$10^{4}k$					120,720	$10^{4}k$
		0.3	6,120	for all				0.3	6,120	for all
			24	$10^{3}k$					24	$10^{3}k$
			720	$10^{3}k,10^{4}k$					720	$10^{3}k,10^{4}k$
		0.5	6,24	for all				0.5	6,24	for all
			120,720	$10^{3}k,10^{4}k$					120,720	$10^{3}k,10^{4}k$
		0.8	6	$10^{2}k,10^{3}k$				0.8	6	$10^{2}k,10^{3}k$
			24,720	for all					24,720	for all
			120	$10^{3}k,10^{4}k$					120	$10^{3}k,10^{4}k$
	Linear	0	6,24,120	for all			Linear	0	6,24,120	for all
			720	$10^{3}k,10^{4}k$					720	$10^{2}k,10^{4}k$
	OAZ	0	6,24	for all			OAZ	0	6,24	for all
$H_{F}$	HaH	0.1	6	$10^{3}k$		$H_{R}^{1/3}$	HaH	0.3	6	$10^{2}k,10^{3}k$
		0.3	6,24	for all				0.5	6	for all
		0.5	6	$10^{2}k,10^{3}k$				0.8	6	$10^{2}k,10^{3}k$
			24	$10^{3}k,10^{4}k$			Linear	0	6	for all
			120	$10^{4}k$			OAZ	0	6	for all
		0.8	6	$10^{2}k,10^{3}k$					24	$10^{2}k$
			24	for all		$H_{R}^{2/3}$	HaH	0.3	6	for all
	Linear	0	6	$10^{3}k,10^{4}k$				0.5	6	for all
			24	$10^{4}k$				0.8	6	$10^{2}k,10^{3}k$
	OAZ	0	6,24	for all			Linear	0	6	for all
$H_{T}^{1/2}$	HaH	0.1	6,24	$10^{3}k,10^{4}k$			OAZ	0	6	$10^{2}k,10^{3}k$
			120,720	$10^{4}k$						
		0.3	6,120	for all						
			24	$10^{3}k$						
			720	$10^{3}k,10^{4}k$						
		0.5	6,24	for all						
			120,720	$10^{3}k,10^{4}k$						
		0.8	6	$10^{2}k,10^{3}k$						
			24	for all						
			120,720	$10^{3}k,10^{4}k$						
	Linear	0	for all	for all						
	OAZ	0	6,24	for all						
			120	$10^{3}k,10^{4}k$						
			720	$10^{4}k$						

**Table 3 entropy-25-00734-t003:** GSS data about religious intolerance.

Year	1998	2008	2018
STRONGLY AGREE	148	285	186
AGREE	429	602	496
NOT AGREE/DISAGREE	278	210	229
DISAGREE	275	196	181
STRONG DISAGREE	72	30	38
Total	1202	1323	1130

**Table 4 entropy-25-00734-t004:** Asymptotic mean and variance of entropies.

	Mean			Variance		
	1998	2008	2018	1998	2008	2018
$H_{S}$	0.914	0.839	0.863	0.0000724	0.0001081	0.0001175
$H_{R}^{1/3}$	0.967	0.933	0.947	0.0001172	0.0002375	0.0002122
$H_{R}^{2/3}$	0.939	0.881	0.902	0.0000317	0.0000500	0.0000512
$H_{T}^{1/2}$	0.932	0.868	0.892	0.0000500	0.0000871	0.0000842
$H_{T}^{3/2}$	0.919	0.849	0.870	0.0000622	0.0001089	0.0001183
$H_{F}$	0.516	0.702	0.642	0.0008028	0.0011355	0.0011917

**Table 5 entropy-25-00734-t005:** *p*-values of the hypothesis of equal entropies.

	1998–2008	1998–2018	2008–2018
$H_{S}$	0.0000000	0.0002606	0.1029
$H_{R}^{1/3}$	0.0705681	0.2520262	0.5316
$H_{R}^{2/3}$	0.0000000	0.0000478	0.0404
$H_{T}^{1/2}$	0.0000000	0.0004471	0.0690
$H_{T}^{3/2}$	0.0000001	0.0002316	0.1672
$H_{F}$	0.0000219	0.0045777	0.2118

**Table 6 entropy-25-00734-t006:** Asymptotic mean and variance of entropies of the collapsed entries of 1998.

	Mean	Variance
$H_{S}$	0.955	0.0000678
$H_{R}^{1/3}$	0.985	0.0000171
$H_{R}^{2/3}$	0.970	0.0000083
$H_{T}^{1/2}$	0.971	0.0000281
$H_{T}^{3/2}$	0.949	0.0000896
$H_{F}$	0.192	0.0000678

**Table 7 entropy-25-00734-t007:** *p*-values of the hypotheses of equal entropies using collapsed data in 1998.

	1998–2008	1998–2018
$H_{S}$	0.00000	0.0000
$H_{R}^{1/3}$	0.00115	0.0107
$H_{R}^{2/3}$	0.00000	0.0000
$H_{T}^{1/2}$	0.00000	0.0000
$H_{T}^{3/2}$	0.00000	0.0000
$H_{F}$	0.00000	0.0000

## Data Availability

Not applicable.

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
