# Peer review of "Asymptotic Distribution of Certain Types of Entropy under the Multinomial Law"

_entropy, 2023, doi:10.3390/e25050734_

Round 1
Reviewer 1 Report
In this paper the authors present expressions for the asymptotic distribution of several information measures.
They ascertain that the Renyi and Tsallis of order q and
Fisher entropies have limit Normal distribution.
This entails variances that depend on the underlying probability of patterns and the number of patterns.
The authors verify that asymptotic distributions are good models for data arising from simulations for a variety of models.
They encounter that the Fisher entropy is the one that fails more frequently to pass the Anderson-Darling normality tests.
The distributions that the authors present can be used for building test statistics and performing tests with mixed types of distributions.
The paper is well written and very interesting,
I recommend acceptance.
Author Response
Thank you very much for your positive assessment.
The authors
Reviewer 2 Report
See the attached document file.

Author Response
Thank you very much for helping us improve the manuscript.
The authors

Round 2
Reviewer 2 Report
I appreciate the authors taking time to address my comments. The manuscript is much improved and the mathematical language is now appropriately precise and clear. I have no further reservations about recommending this paper for publication.